# Gender discrimination in Swedish family courts: A quantitative vignette study

**Leonard Ngaosuvan[1], Jenny Hagberg[2], Sverker Sikström[2]\***

**1** Department of Culture and Society (IKOS), Division of Social Work (SOCARB), Linköping University, Linköping, Sweden, **2** Department of psychology, Lund University, Lund, Sweden

\* sverker.sikstrom@psy.lu.se

## Abstract

### Background

Gender discrimination of women is often emphasized in work contexts, whereas less focus is on how men are discriminated against in social relationships. Gender discrimination in decisions of family relations, is essential to study as the contact between parent and child is commonly viewed as the most important relationship in people's life, as well as being the most important aspect of our life. Following separations, decisions on custody disputes are made by social workers. The purpose of this paper is to study gender discrimination in such decisions.

### Method

Participants were instructed to give a recommendation of shared parenting based on a custody case vinjett, where we manipulate the gender of the risk parent.

### Results

The participants' recommendation of living was mainly dependent on the risk parent's gender, where the mother was considerably more likely to receive shared custody than the father.

### Conclusions

Professional social workers show selective gender discrimination against fathers in terms of living recommendations.

## Introduction

Gender discrimination has been thoroughly investigated in several academic fields such as social cognition, psychology, law, social work, political science and gender studies, where the dominating focus is on how women are discriminated against in the public domain. A meta-review of the last two decades of gender equality research used computational methods to

**Data Availability Statement:** Data is avaliable at: https://osf.io/ksdbn/?view_only=59afacdf58f34da68238f156e932a2e7.

**Funding:** The authors (SS and LN) received a grant from FORTE (2023-01833). The funder did not play

any role in the study design, data collection and analysis, decision to publish, or preparation of the manuscript.

**Competing interests:** The authors works with investigations of custody cases part time.

summarize 15465 scientific articles into 27 main topics [1], where almost all of these topics emphasized discrimination against women in the public domain. At the same time, when people are asked what domains are important in their lives, the overwhelming majority of both men and women emphasize that the private domain, in particular social interaction in family, is the most important aspect in their lives [2]. Power, in this social interaction between children and parents, are primarily decided by social workers as parents divorce, following a custody dispute. In this vein, there is a remarkable lack of studies on gender discrimination in custody evaluation recommendations.

Due to the experienced importance of such custody decisions, it is essential to investigate whether gender biases can be identified in this domain. To investigate this topic becomes even more important as there are few randomized controlled studies in this topic (for an older exception, see [3]). Here we first review the debate on gender discrimination in child custody recommendation, followed by a brief review on studies of gender discrimination in custody cases. Finally, a vignette study is conducted where social workers make custody evaluations where the gender of the risk parents is subject to randomized controlled manipulations. In our review of prevailing conditions, we focus on the Swedish contexts as the data was collected in this country.

## Custody disputes in Sweden

The Swedish legal process in child custody is similar to many other Western countries. In short, a custody dispute begins after mediation has collapsed and either parent is suing the other for child custody. Then, the court orders a custody evaluation from the family court unit at the social service. In Sweden, social workers are conducting custody evaluations. These custody evaluations are subsequently sent to the courts where court hearings are held, and final decisions are ruled. This system creates a potential diffusion of responsibility between the family court (social service) and the judicial courts. However, in practice, similar to international findings [4] judicial courts rarely go against the recommendations from the social services in Sweden [5].

In Sweden, a reform was enforced in 2006, where conflict between parents became a legal justification for sole custody, thus creating a motivation to magnify a conflict as it improves chances of winning sole custody. Following this reform, the number of custody cases doubled, and has henceforth stabilized to around 7 000 cases per year.

## The gender discrimination debate

There are two opposing sides in the discussion of gender discrimination in custody cases, where one position is that women are discriminated against as mens' physical *violence* are ignored, whereas the other view suggests that *conflict* arises as men are discriminated against due to that custody evaluators have a negative preconceived conceptions of fathers' parental skills [6]. Proponents of the violence explanation, including researchers (e.g., [7]) and women's rights' organizations, emphasize violence towards parents or children. That is, women are discriminated against because mens' abuse is not sufficiently considered because of lack of physical evidence (i.e., convictions in criminal courts) in child custody cases. In contrast, proponents of the conflict explanation argue that false accusations is a strategy for women to win custody disputes (see [8] for a discussion on false allegations after separations). Furthermore, they claim that fathers are being discriminated against because of negative preconceived conceptions of fathers, which is emphasized by an ingroup bias where the majority of custody evaluators are women.

The validity of the violence explanations can be tested empirically as it is based on the assumption that custody disputes largely depend on violence. Swedish Gender Equality Agency [9] investigated the proportions of violence in custody disputes in 2021. The violence explanation was partially supported because criminal convictions of violence were present in 20% of the cases. The conflict explanation was also supported because there were no convictions or accusations of violence in 36%. In the remaining 44%, there were accusations of violence without sufficient evidence leading to criminal convictions.

Another way of reconciling the violence versus conflict views is that they are based on different definitions of violence. Some municipalities use the definition: "Violence is any act directed against another person that through this action scares, causes injury, pain or that gets that person to do something against their will or to stop doing something that they want."(Isdal, 2001), which opens up the possibility of viewing virtually every high-conflict separation as violence. Another definition of violence is that "any behavior within an intimate relationship that causes physical, psychological, or sexual harm to those in the relationship" [10]. Thus, whether fathers or mothers are being discriminated against may simply depend on the choice of definitions of violence, turning the debate more into political rhetoric, rather than science.

### Research on gender discrimination

A more meaningful approach to understand gender discrimination in custody cases is to look at empirical studies. Here we look at register, legal sociology and vignette based studies.

### Register studies

Register analyses can be used to understand gender discrimination. About 79% of all US parents with sole custody are mothers [11], which is not necessarily the same as court rulings, as a large proportion of these cases are settled by an agreement of sole custody for the mother. In the remaining 28% of the cases the mother won against the fathers will. Hence, such statistics need to be understood in the context of confounding factors. For this reason, register studies are informative, but often lack strong internal validity.

### Legal sociology and survey

Legal sociology studies involve readings and interpretation of court rulings. Schiratzki [12] investigated gender discrimination in Swedish custody disputes in rulings from the Court of Appeal and claimed that positionality, stability and previous engagement with the child were deciding factors rather than parental gender. However, the legal sociology method has flaws that make it practically difficult to detect gender discrimination due to three reasons. First, an evaluator can, and is also likely to, exhibit gender discrimination without being aware of it. Thus, judges may unknowlingly discriminate against men. Consequently, they will not, or simply are not able, to spell this out in the rulings. Second, in Sweden it is illegal to discriminate against either gender. As judges are very well informed about the law, it is highly unlikely that they would incriminate themselves by ruling in favor of one parent because of their gender. Third, even if gender discrimination was legal in Sweden, it is not socially accepted. People often behave according to social desirability (e.g., [13]), which is particularly true when they know that they are under surveillance, and court rulings are highly scrutinized documents. For these reasons, legal sociology studies are not the most suitable scientific method to detect gender discrimination in custody disputes. Social desirability is also very prominent in surveys (e.g., [14]). For instance, judges subjectively report (15%) far less favoritism towards mothers compared to attorneys (49.9%), which is not surprising [15].

## Vignette methodology

A vignette study is a powerful method to measure discrimination because it allows for randomized controlled experiments with strong internal validity. Traditionally, vignettes use between-subjects designs where participants are instructed to rate, judge or suggest something based on the information presented by the researcher. For instance, participants could be instructed to read a story about a person in need, and asked about their hypothetical actions (e.g., [16]). In one condition, the person in need is a man, and in the other condition the person is a woman. If there are differences in reported behavior then researchers can conclude that gender discrimination is evident. Vignette methodology has previously been used to study child custody. For instance, Ngaosuvan et al [17] used a within-subjects design where participants read custody dispute vignettes and rated how cooperative they would have behaved. Costa et al. [18] used the vignettes method to show that gender stereotypical attributes such as being friendly, generous and trustworthy were imposed by the participants as female traits and showed how this could explain gender differences in child custody cases. Kullberg [19] showed that Swedish social workers assessed single mothers and fathers selectively in two principal ways. Single fathers were considered as having more severe issues compared to mothers and were assessed as being less deserving of help than single mothers. Although the study was not in a child custody context, the results are in line with gender discrimination favoring mothers over fathers. A similar vignette study [20] found that a representative sample of participants assigned more postdivorce custody to mothers when gender roles were traditional. Sagi and Dvir [3] found that female custody evaluators discriminate against men in custody evaluation recommendations. The above vignette studies on gender discrimination in parenting or custody disputes, have consistently shown that men are being discriminated against. However, there are several limitations of previous research in vignette studies on gender discrimination in custody disputes. First, the cited studies are dated (e.g., [3]). This limitation is very important because gender equality measures not including custody evaluations are constantly improving in many nations, and there are reasons to suspect that improvements have taken place. Second, the World Economy Forum's Global Gender Gap Index (GGGI) puts Sweden at .81, which is at the sixth top country in the world. In comparison, Israel, where Sagi & Dvir's [3] study took place, GGGI is .71, which is today in the middle of countries worldwide. Thus, in Sweden being 'one of the most gender equal nations' in the world, gender discrimination against men in custody disputes are of particular interest because many reforms for increased gender equality have been aimed at improving the lives of women while, presumably, neglecting men's perspective. We only found one relevant study using Swedish social workers [19], but it did not contain recommendations on child custody, living or visitation. Third, despite the extensive and expanding field of gender discrimination, studies on gender discrimination in custody evaluations with professionals as participants have been sparse. This prompts an important research knowledge gap: In relatively gender equal nations such as Sweden, how gender equal are family court recommendations in custody disputes among its active professionals?

## Historical overview

From a more historical perspective, there are several principles that could describe how custody disputes have been resolved, namely, (a) the tender years doctrine, (b) the primary caretaker preference, and (c) the psychological parent [21].

## The tender years doctrine

Initially created to be gender-neutral, this doctrine stipulated that custody should be awarded to the parent who has carried out the caring and nurturing tasks of parenting such as preparing

meals, bathing, grooming, dressing as well as teaching the child elementary scholastic skills such as reading, writing, drawing and simple math. For quite some time, the tender years doctrine was used to favor mothers in child custody cases [6, 22].

## The primary care-taker preference

This normative theory assumes that the parent who has exercised most parental responsibility should be awarded sole custody. As Kaminska [21] puts it: "The primary care-taker was the parent who handled all or most of the daily tasks related to the child, including grooming, dressing, eating, cleaning and household chores, shopping or managing medications."

Historically and traditionally, women have done these tasks more than men. This is particularly true when the child is an infant or very young. However, the importance of fathers becomes more pronounced as the child grows older, are more ready to be introduced to adulthood and how to handle the outside world. However, there is a strong notion that mothers are the primary caregiver in many cultures. As Warshak [6] points out, there are several care-taking activities associated with traditional female gender stereotypes that are focused on in custody disputes. In contrast, such activities are typically not associated with male gender stereotypes. Similar to the tender years doctrine, the notion of the primary care-taker favored mothers over fathers.

## The psychological parent

This theory was introduced by Goldstein, Freud and Solnit [23]. The core assumptions are that; joint custody is disruptive for the child, that there should only be one custodial parent, and the other parent could only visit the child at the custodial parent's discretion. The reasoning was that it is better to keep one relation with one parent rather than losing relations to both parents. The psychological parent is the parent who is closest to the child. Following the same pattern as the tender years doctrine and the primary care-taker preference, the idea was initially advertised as gender-neutral, but in practice, the psychological parent had the same effect: Mothers were favored.

## The child's best

This principle is the most commonly used guideline to resolve custody disputes in contemporary Sweden. However, there are several conceptual problems with the principle. Since the concept of the child's best is undefined, the principle lacks instructions for custody disputes. Swedish expert Governmental Agencies such as the Family Law and Parental Support Authority [24] who is responsible for these issues simply refuses to explain what the child's best actually means: "Concepts like child perspective, the child's needs and the child best are concepts that are hard to define. The concepts are not singular and their meanings vary depending on individual, situation and context. [–––]. What is best for the child must be determined in each individual case from an assessment of the individual conditions." This refusal to define what the concept means combined with an acceptance of case by case assessments, in practice provides a motivation to support gender discrimination. These two conditions allow discrimination in several ways. First, professionals can use whatever meaning they choose in each case, even meanings that explicitly or implicitly favor mothers without repercussions. Second, professionals could argue that each case should be determined on its individual situation and context and thereby never accept comparisons with other cases. This virtually eliminates any possibility to prove that discrimination has taken place. This is very important because gender discrimination is illegal in Sweden but without any plausible strategy to provide meaningful evidence, the discrimination law becomes a paper tiger. Third, lack of priorities also helps

discrimination to thrive. As pointed out by [25], discrimination can occur even with generally established conditions without priorities. Assume that a father is in two custody disputes with two different mothers in two different municipalities. There are only two relevant family conditions, A and B. In one case, the father has more A, but less B. In the other case, he has more B but less A. However, the custody evaluators could recommend sole custody for the mothers in both cases because in the first case B was considered the most important condition for the child's best and in the second case A was the important condition.

## Significance of gender discrimination in child custody

In Sweden, the feminist movement has made great strides in the last fifty years. For example, women are now dominating the educational system; girls have had higher school grades for 22 years running [26], there are more female University students, and there are lower demands to reach professorship for female researchers [27]. While the Swedish feminist movement has been successful and created a more gender equal professional domain, the female-dominant private domain (e.g., decision power about homes and family) has been left unchanged (see [2]). Indeed, in Sweden women can inseminate themselves with the support of the Government, while surrogate mothering is illegal. Furthermore, female dominance in family decisions is also evident as men (particularly those with low income) suffer from involuntary childlessness to a higher degree than women [28]. In fact, female privilege concerning dating, courtship and family creation is so strong that researchers suggested a sexual-economic market theory based on it [29]. Thus, gender discrimination against men in custody disputes is essential to study in order to assess the magnitude of female dominance in the field.

## Aims and hypothesis

The aim of the current paper is to use vignette methods to investigate gender discrimination in social workers custody case recommendation. Based on previous vignette studies on gender discrimination in custody disputes [3, 18, 20], and historic female favoritism in custody disputes, we expected that social workers would recommend in favor of mothers.

## Method

### Design and participants

The study comprised a simple experimental vignette design with parental gender as independent variable, and recommendations of custody, living and visitation as dependent variables. Data on risk was collected but will be reported in another study, and is not focused here (see, [30]). Participants were recruited from municipalities in the midwest of Sweden, including the Stockholm area. A total of 29 Swedish custody evaluators (8 men and 21 women) participated in the study. This skewness represents the gender distribution in the field fairly well as the entire profession consists of 86% women [31]. The sample was fairly experienced as family court evaluators (M = 6.44 years of experience, SD = 6.36).

### Materials

A vignette story of a custody dispute was developed for the study depicting one parent (father or mother) as weak and the other parent as fully functional (see Supplementary Material). There were two versions of this vignette, that were identical in all aspects except for the gender of the weak and the functional parents. The weak parent was described as more strict than the other parent, who believed that the child's hair had been pulled several times. Also, the weak parent forgot to pick up the child from kindergarten five times, and had a history of

depression. Presently, the weak parent was feeling better, but still used psychopharmaceutical medicine and had semi-regular contact with a psychologist. To ensure quality of the vignette, an experienced custody evaluator (5+ years of experience) was consulted. The child expressed a wish to live with the functional parent. The functional parent was described as having enough and sued the weak parent for sole custody. In total, the vignette was 567 words long. Risk assessments were assessed by crossing an X on a 10 cm line from minimal risk to the left, and maximum risk to the right. Recommendations were measured in one multiple choice question with the following alternatives: A. Sole custody and living for the functional parent, and limited visitation for the weak parent. B. Sole custody and living for the functional parent, and extensive visitation for the weak parent. C. Shared custody, and shared living. D. Shared custody, while the functional parent is awarded living, and the weak parent is awarded limited visitation.

## Procedure

First, participants were informed that the study's purpose was risk assessment in custody evaluations. They were then given informed consent to participate in the study. Then, the participants were asked about demographic information (years of experience in social work, years of experience as family court evaluator, gender and age). After that, the participants were given definitions of physical violence, psychological neglect, physical neglect, and abduction. Then, participants were randomly assigned to either the weak father or weak mother experimental conditions. Participants in the weak father condition read an abbreviated version of a custody evaluation with a general background of a present situation and a description of the father as moderately weak as a parent. Participants in the weak mother condition read the same version except that the mother was the moderately weak parent. Then, participants rated subjective levels of risk for each type of maltreatment that they were given a definition of. Finally, participants were asked to choose recommendations for a subsequent court ruling.

*Ethics*: The methods and procedures in the study were conducted in adherence to the relevant Helsinki Declaration ethical principles. According to Swedish law and the Swedish Ethical Review Authority, this study does not need to be reviewed, as it does not include: identifiable/ sensitive information, a physical manipulation, biological material or a potential risk for the participants. Subjects were informed of the right to refuse to participate in the study, or to withdraw to participate in the study at any time without reprisal. To participate they were given written informed consent. The data was collected anonymously and did not include identifiable material.

## Results

Group comparisons using between-group simple ANOVA showed no differences in participant total working experience (Weak mother; M = 23.00, SD = 6.96) vs Weak father (M = 18.00; SD = 9.32), or working experience as a custody evaluator (Weak mother; M = 7.10, SD = 6.38) vs Weak father (M = 4.53; SD = 3.98). However, there was a tendency that the weak mother condition participants (M = 53.31, SD = 8.23) were slightly older than the weak father condition participants (M = 46.69, SD = 9.13), F(1, 28) = 4.12, p = .052.

Recommendations of custody, living and visitation are related in a complex manner. For instance, it is very common that parents with sole custody also are the principal living parents, parents with living rights cannot have visitation rights, and participants cannot recommend both sole and shared custody. However, custody evaluators and participants in the present study can recommend shared custody but not shared living. Thus, there are three ways gender discrimination can be revealed in the present study. First, participants can recommend sole

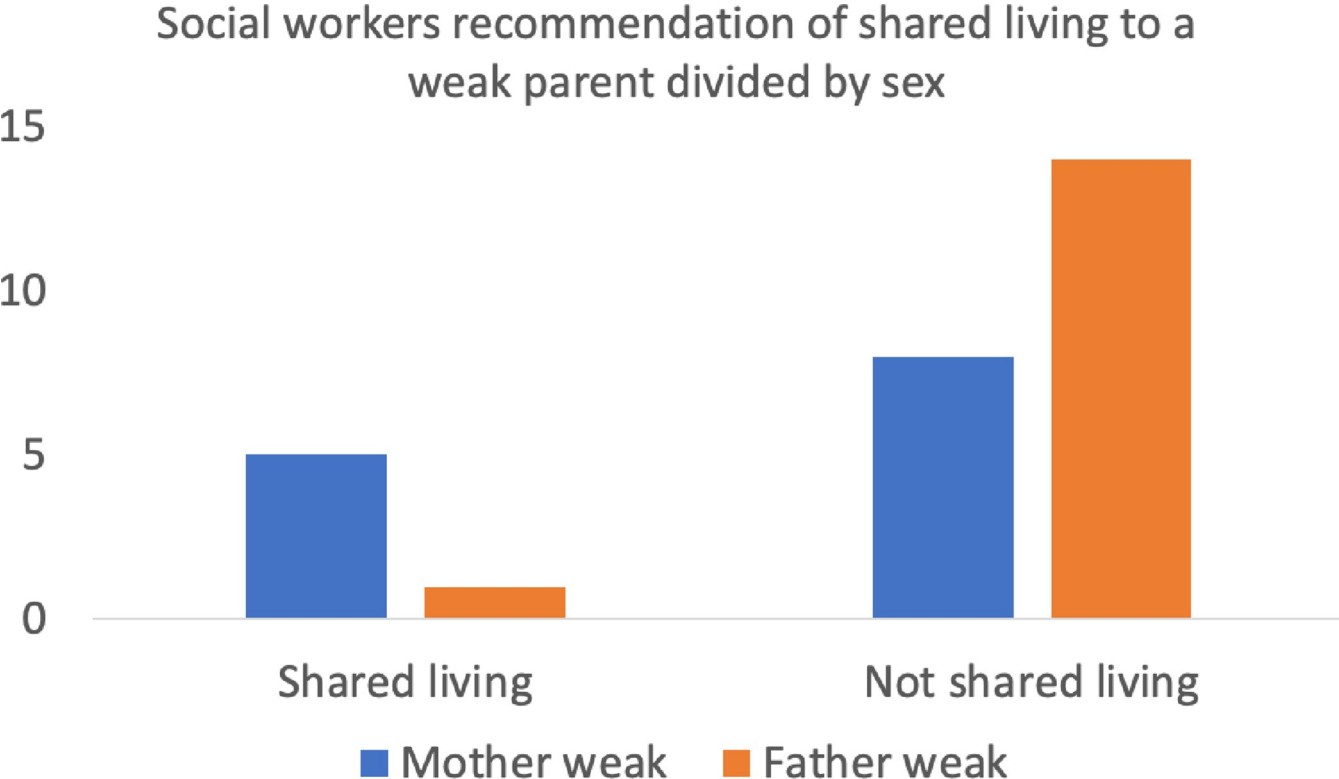

**Fig 1.** Shows the number of social workers recommending (y-axis) shared living (left) or not shared living (right), divided into when the mother (blue) or father (red) is portrayed as weak.

custody for mothers to a higher degree. Second, participants can recommend unequal time living with the child. That is, both parents are recommended shared custody, but mothers are to a higher degree recommended shared living than fathers. Third, recommendations of visitation rights can also discriminate against fathers by either recommending more visitation for mothers than fathers, or simply recommending no visitation at all for fathers more often than mothers. Thus, depending on the preferences of the participants, gender discrimination can be evident in either custody, living or visitation.

One participant did not provide a recommendation. There were too few sole custody recommendations (6) for statistical analysis. Functional fathers were recommended sole custody by two participants and functional mothers by four participants. The participants showed gender discrimination in terms of ratio recommended shared and not shared living (see Fig 1). For weak fathers the ratio was 1 (shared) to 14 (Not shared). In comparison, the participants recommended a ratio of 5 to 8 for weak mothers. This difference was statistically significant, $\chi^2$ (1, 28) = 4.12, p = .04 < .05.

## Discussion

The present study investigated gender discrimination in a hypothetical custody dispute using a vignette methodology. The results showed support for the gender discrimination hypothesis in recommendations for living. However, shared living was recommended to mothers to a higher degree compared to fathers. That is, fathers with the identical weaknesses as a mother are less likely to be recommended for shared living compared to mothers. This result is in line with previous international studies [3, 18, 20].

The results can be interpreted in several ways; namely, (a) elegant gender discrimination against fathers, (b) low-conflict separation, and (c) discriminatory child perspective against fathers. *Elegant gender discrimination* against fathers suggests that the participants are aware that parents who are recommended for sole living often also are awarded sole custody in court. In Sweden, courts usually do not rule shared custody but sole living with one parent. Presumably, the participants being experienced custody evaluators are fully aware of this. So, if participants implicitly favored mothers, they can choose to recommend favorably for mothers in terms of living only knowing that the courts, who make the final rulings, probably would rule for sole custody and living for the functional mother. Another explanation is that the participants understood the separation as *low-conflict*, and therefore recommending for sole custody is not necessary. However, despite the fact that the weaknesses were identical for both parents it only had an effect on fathers, as they were more likely to be recommended to be excluded from the child's life than mothers. *Child perspective* is important in all custody disputes, and could have guided the participants to follow the child's wishes. In the vignette, the six-year-old child expressed a wish to live with the functional parent. However, the results indicated a gender bias in following the child's wish. Only, when the child wished to live with the mother, the wish was relevant for recommendations. Regardless of the explanation, the results are similar to previous vignette studies [3, 18, 20] but in a Swedish context.

A possible reason for that gender discrimination was evident in living, but not custody, is case severity. The vignette concerned a mild parental conflict with relatively small weaknesses in the weak parent. This explains why so many participants recommended shared custody, and subsequently gender discrimination was evident in living recommendations. In a more severe conflict, where it is quite obvious that sole custody is the best option, gender discrimination would probably be more evident in visitation rights. That is, participants would recommend weak mothers more visitation rights than weak fathers.

## Weaknesses

In the present study, we have identified the following weaknesses; (a) ecological validity, (b) insufficient information, (c) limited time for consideration, (d) lack of measurement of vignette quality and realism, (e) partial process measurement, (f) task complexity, (g) legal system correction, and (h) limited power.

The major drawback for many vignette studies is *ecological* validity. The current study does not reflect the working situation, or the richness of a natural custody evaluation. For instance, the participants in the present study had *limited information* as they did not meet the parents or the child, nor have they visited their homes. Furthermore, the decision was individual as opposed to the collective decision-making that occurs in real cases.These aspects make the research situation different from the natural setting, where decisions on recommendations are collective efforts that are discussed with colleagues. Furthermore, a traditional custody evaluation takes *considerable time* which allows for reflection and careful consideration. In the present study, the participants read a short summary and then answered questions and made a recommendation in short order. This is not how traditional recommendations in custody disputes are decided.

The present study did not include participants' subjective ratings on vignette quality and realism. This is important because an overly unrealistic vignette story about two custody-disputing parents may disrupt the integrity of the vignette. However, the vignette was designed to represent a mild conflict with realistic parental weaknesses for both genders, and the vignette was developed with the help of a very experienced custody evaluator.

Another potential limitation of the study is that the legal system may correct for gender discrimination. The social services only provide recommendations for the courts. As such, it is possible that the courts reduce the risk of gender discrimination in custody evaluations in their rulings. Social work is an extremely female dominated field as 86% are women [31]. In comparison, legal professionals are more balanced (59%), and only 52% of jurors are women [32]. However, this notion is counteracted by studies showing that courts often rule in accordance to custody recommendations internationally [4] as well as in Swedish contexts [5].

The study used a small sample size which is a clear limitation in terms of statistical power. However, the fact that we found significant results based on a small number of participants actually suggests a very strong effect size of discrimination. Previous, albeit international [18, 20], dated [3] or related to parenting [19] research all point to the notion of gender discrimination against men.

As the legal system of custody disputes in Sweden concerns two different authorities; social services (family court) and judicial courts, the present vignette only investigated custody evaluators' recommendations, and not court decisions, there is a risk that additional discrimination may occur in the judicial court. However, courts usually follow the recommendations of the social service (family court) recommendations [5], they are the most essential pieces of evidence in decisions for custody disputes.

In terms of the gender debate, the present study vignette did not contain any type of accusations of intimate partner violence. This is interesting as the results show gender discrimination against men in a custody dispute even where there are no indications of IPV. This finding is not supported by the violence account as it would argue that it is the fathers' violence per se that drives the gender discrimination. Furthermore, social service recommendations and court rulings may show different gender biases in terms of violence. This line of argumentation leads to a third hypothetical answer to the debate where both genders are discriminated against, men in social service custody evaluation recommendations concerning living, whereas women are discriminated against in the court rulings. Thus, both explanations may be correct, but in different agencies' exercises of authority.

Task complexity is probably critical for the detection of gender discrimination in custody disputes. In the present study, the conflict and weaknesses were relatively mild. In such cases, the probability of finding gender discrimination is larger compared to more clear cases. If the case is very obvious that the weak parent in the vignette is unfit, then no gender discrimination should be evident. This informational constraint is very important when considering gender discrimination. That is, the complexity of the cases will probably affect the levels of gender discrimination.

## Implications for practitioners

Despite the small sample size and weaknesses, the results are in line with international research on gender discrimination, and should not be regarded as surprising. For practitioners, the present study could be viewed as a wake-up call for professional custody evaluators that men are at risk to be discriminated against in custody evaluation recommendations. This first step for professionals would be to admit that there is a risk. Then, there are, at least, two main strategies that could alleviate this problem, namely, (a) cultural changes, or (b) methodological changes. Cultural changes concern changing the cultural view on gender and parenting. Societies, such as Sweden, that actively strive to be gender equal in other areas should consider developing a gender-neutral culture in the parenting domain as well. However, as the aforementioned historic overview shows that the same problems with gender-neutrality are evident in several different methodological approaches in custody evaluations, such as the tender years

doctrine, the primary care-taker, the psychological parent, and the childs best. This may be particularly difficult to deal with as the common problem these approaches are gender stereotyping where Kaminska [21] correctly notes: "As it has turned out, stereotypes die hard." The second strategy would be to use a method that reduces the risk or gender discrimination. One such method has been proposed by Ngaosuvan [33] namely anonymization by using two groups of evaluators. Ngaosuvan suggested that the first group of custody would conduct the data collection, and write an anonymized report where most aspects subjected to discrimination (gender, age, profession, lifestyle and so on) is masked. Then, a second group of evaluators, preferably stationed on a different location, would read, discuss and then provide the recommendation.

## Implications for fathers and children

Being discriminated against is frustrating, and usually, the ones suffering the most are the ones that are directly affected by the decisions. For fathers, there is a finality in family courts because there are no other venues to go to. Perhaps more importantly, discriminating against fathers in custody causes great harm to affected children who will grow up with limited, or no, contact with their father. In a broader perspective, the introduction briefly summarized that women dominate courtship and family formation [2, 29]. To what extent discrimination against fathers is caused by the fact that virtually every aspect of family and children are dominated by women needs to be further investigated.

## Future studies

Future studies may benefit from investigating a similar methodology but with added accusations of violence in the vignette. This would investigate the violence explanation. Furthermore, the present sample is too small to analyze gender differences in participating social workers. For instance, it would be natural to investigate whether female professionals discriminate against fathers to a higher degree than male professionals. In addition, other discriminatory factors such as ethnicity, race and culture using the vignette methodology would be an interesting line of future studies (see [34]).

## Conclusions

The present paper shows gender discrimination in custody evaluations recommendations against men in vignettes made by social workers. Further studies should look into the extent, where and how gender discrimination occurs in real custody disputes. For instance, the present study can be replicated but with an authentic and complete custody evaluation as stimulus materials.

## Supporting information

**S1 File.**
(PDF)

## Acknowledgments

We thank Jukka Kaski and Elin Tegnér for assisting in data collection. This paper is based on data collected for their Bachelor's thesis in social work with their permission.

## Author Contributions

**Conceptualization:** Leonard Ngaosuvan.

**Data curation:** Leonard Ngaosuvan.

**Formal analysis:** Sverker Sikström.

**Investigation:** Leonard Ngaosuvan.

**Methodology:** Leonard Ngaosuvan.

**Project administration:** Leonard Ngaosuvan.

**Supervision:** Leonard Ngaosuvan.

**Validation:** Leonard Ngaosuvan.

**Writing – original draft:** Leonard Ngaosuvan, Sverker Sikström.

**Writing – review & editing:** Leonard Ngaosuvan, Jenny Hagberg, Sverker Sikström.

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
