## [Decision Letter · Decision Letter 0]

11 Mar 2024

PONE-D-23-40878Gender Discrimination in Swedish Family CourtsPLOS ONE

Dear Dr. Sikström,

Thank you for submitting your manuscript to PLOS ONE. After careful consideration, we feel that it has merit but does not fully meet PLOS ONE’s publication criteria as it currently stands. Therefore, we invite you to submit a revised version of the manuscript that addresses the points raised during the review process. Both methodological and structural implementations are needed. Revisions are needed to improve clarity and depth.

We look forward to receiving your revised manuscript.

Kind regards,

Andrea Cioffi

Academic Editor

PLOS ONE

[The authors works with investigations of custody cases part time.]. 

3. In the online submission form, you indicated that [Relevant data are within the manuscript. Additional data can be obatined by emailing the corresponding author]. 

Reviewers' comments:

Reviewer's Responses to Questions

**Comments to the Author**

1. Is the manuscript technically sound, and do the data support the conclusions?

Reviewer #1: Partly

Reviewer #2: Yes

Reviewer #3: Partly

Reviewer #4: Partly

2. Has the statistical analysis been performed appropriately and rigorously? 

Reviewer #1: No

Reviewer #2: Yes

Reviewer #3: No

Reviewer #4: No

3. Have the authors made all data underlying the findings in their manuscript fully available?

Reviewer #1: No

Reviewer #2: Yes

Reviewer #3: No

Reviewer #4: No

4. Is the manuscript presented in an intelligible fashion and written in standard English?

Reviewer #1: Yes

Reviewer #2: Yes

Reviewer #3: No

Reviewer #4: Yes

5. Review Comments to the Author

Reviewer #1: Review PONE-D-23-40878 – Gender Discrimination in Swedish Family Courts

This paper provides insights into possible gender discrimination in decisions about custody decisions. Using a vignette approach with Swedish social workers, this paper aims to provide more insights into possible gender discrimination, specifically against fathers, in such decisions.

Bias research in the legal domain is an important matter of study, especially in such important cases as custody cases. The choice for a vignette design explained properly, and the sample with professionals makes it a unique study. However, I believe this paper needs revisions before it would be suitable for publication in all sections of the paper.

First, I believe the introduction can be structured in a clearer way. I appreciate the explanation of the Swedish system as this is relevant for the paper, but the theoretical framework and previous literature can be structured in a better way. What are exactly the limitations of previous research, and how does this study fill these gaps?

Furthermore, the only relevant hypothesis, as I understand from the paper, is the “female primary caregiver hypothesis.” However, the only source to explain this hypothesis is more than 25 years old. If possible, I believe it would greatly increase the strength of the theoretical framework if more recent literature can be incorporated, especially empirical studies on this. It would be even stronger if there is any literature based on a Swedish sample.

Second, the methods section needs to be improved. Although the design and procedure are explained, more detail on the sample and groups are worth stating. What are the group descriptives, were there any group differences? The sample size is very low, which is understandable with a sample of professionals, but what does this mean for the power of the study?

I believe most revisions should be done in the results and discussion section. The result section now only shows statistics for the found difference in recommendations of shared living, but there are more options to answer. For completion, it would be more transparent to also show the non-significant results. Especially because the discussion section does try to explain some of the non-findings, but no statistics are given. The result section should also include some descriptive statistics of the answers, as for example in the discussion the scored weakness of the parents is discussed, but no statistics are provided.

Furthermore, research on bias is important but before you can conclude that there is a bias, it is important to exclude other possible explanations of differences. Now, nothing is stated about any additional analyses trying to find other explanations such as gender of the participant or years of experience. If this is not possible with the sample size, the authors should consider trying to increase the sample size.

In general, the result and discussion sections require more explanation about made choices for the analysis. The method section mentions four answer options, but the structure and text of the result section makes it difficult to follow which answer options are analyzed or taken together. The second paragraph of the discussion was very hard to follow, I believe the first sentence should be “… sole living but not sole custody…” instead of shared custody.

The discussion section discusses some possible explanations of the findings but does not elaborate on the non-finding for custody vs. the finding on living. It would strengthen the paper if the authors could come up with an explanation for this as this is somewhat surprising for me as a reader.

What I am missing in the limitation section is the limited sample size. I understand how difficult a professional sample is, but I believe this is an important limitation of this study as it greatly limits the generalizability of this study. The authors should be careful with making strong conclusions based on this small sample size. This would also be an important suggestion for future research.

A minor thing, but I hope that there is more recent work on unrealistic vignette stories than from 200 years ago.

In conclusion, I believe that the paper holds valuable information about gender discrimination in courts, with a unique sample. However, I believe revisions are needed to increase the quality of this paper, especially providing more statistical information, a better structure in introduction and discussion as the paper is somewhat hard to follow, and more nuance with the conclusion as a result of the small sample size.

I hope these comments may be helpful to the authors. Best wishes!

Reviewer #2: It is fantastic to see research on the impact of the legal system on men. There is definitely a gap in the literature that needs to be filled. While the paper focuses on an important topic, it would be good to further examine the impact of your findings. What does this mean for practitioners? What needs to happen in the field to reduce the negative impact on men? It might be worth looking into therapeutic jurisprudence and examine the impact these decisions may have on men in the short and long term. By weaving legal theory and discussing how your research translates to practice you will create a stronger paper. There are also quite a number of spelling and grammatical errors throughout the paper that need to be addressed.

Reviewer #3: In a vignette study, the authors found that mothers were considerably more likely to receive shared custody than fathers.

The authors asked participants to provide a risk assessment as well as to make a living decision. However, they only focus on the living decision and do not report the risk data.

Due to the very small sample size and the relevance of the research questions, I recommend reporting and analyzing the risk data here. This would enhance the contribution of the paper and allow readers to assess the entire decision-making process.

It is unclear from the manuscript whether the risk question was asked before the living recommendation. There is concern that the order of the questions might affect the responses, so perhaps a randomized order would have been the best strategy. I also recommend conducting a mediating analysis.

The sample size is very small: a total of 29 Swedish custody evaluators (8 men and 21 women). Relatedly, was there an interaction between the gender of the decision-maker and the gender of the parent? I am concerned that the sample size is too small to investigate this adequately.

The authors acknowledge the risk of additional discrimination occurring in the judicial court. However, it could also be that courts "correct" the discriminatory decisions of evaluators. This should also be stated as a limitation.

Contribution: There are some studies on gender discrimination in custody disputes. Although dated, Sagi and Dvir (1993) had already shown that female custody evaluators discriminate against men in custody evaluation recommendations. I reiterate my point about the risk data and why I believe it should be discussed in this paper, as it would enhance the contribution.

Reviewer #4: Overall, the manuscript indicates the exciting point of 'discrimination against men in custody dispute'. However, it

requires more details on the significance of the study, its methodology and results for meaningful discussions.

The manuscript's introduction still lacks an explanation of the importance of gender discrimination in men's studies.

It is still being determined whether the research is quantitative or qualitative.

As the quantitative research, the sampling technique should be demonstrated. From which population can a representative samples of 29 social workers be resembled?

As for the qualitative research, the data results are invalid for hypothesis testing. The demographic data of all social workers should be included in discussions. Previous work on gender discrimination with the vignette method demonstrates that both the gender and social capital of participants play a significant role as well.

Instead of discussion on the 'weaknesses' part of the study, the study should close the gap in its methodology and provide more insightful findings for further discussions.

6. PLOS authors have the option to publish the peer review history of their article (what does this mean?). If published, this will include your full peer review and any attached files.

Reviewer #1: No

Reviewer #2: No

Reviewer #3: No

Reviewer #4: No

---

## [Author Response · Author response to Decision Letter 0]

8 May 2024

Rebuttal letter

First, we like to thank all reviewers for your useful comments and suggestions. We believe that your valuable comments have substantially improved the manuscript. With few exceptions, we have revised the manuscript to address your insightful and important comments. In addition, we have consulted a legal expert who graciously has accepted being a co-author to improve the overall quality and, in particular, the manuscript's legal terminology. 

Responses to the reviewer's comments are listed below. 

Reviewer #1

Introduction

1. Introduction can be structured in a clearer way.

Response: We have revised the introduction and added a more historical perspective on why mothers are favored. Also, we have added a section, along with a more clear explanation gap: Previous research is either (a) old, (b) outside of Sweden, or (c) using professional social workers as participants. The present study is the only study that satisfies all three conditions.

Theoretical framework

The strength of theoretical framework if more recent literature can be incorporated, especially empirical studies on this. It would be even stronger if there is any literature based on a Swedish sample.

Response: We have broadened the literature search and added more recent relevant Swedish research in Kullberg (2005). Although his vignette study is not directly concerning custody disputes, it suggests favoritism for mothers in parenting. However, as much as we would like to add more recent Swedish research in the field, the number of active Swedish researchers interested in custody disputes are very few. The first author know them by name, and none of them are interested in gender discrimination against men. In fact, as far as we know, the only researchers interested in gender discrimination in Swedish child custody are the authors of this manuscript.

Methods section I

Second, the methods section needs to be improved. Although the design and procedure are explained, more detail on the sample and groups are worth stating. What are the group descriptives, were there any group differences? The sample size is very low, which is understandable with a sample of professionals, but what does this mean for the power of the study?

Response: It means that the effect size (i.e., the gender discrimination) needs to be very high for the study to find significant effects. This is indeed the case. We have added group comparisons in the Design and Participants section.

Methods section II

The sample size is very low, which is understandable with a sample of professionals, but what does this mean for the power of the study?

Response: We have added limited power as a listed weakness in the study, as well as discussing that the effect size could be very strong because of the small data set.

Results and discussion section I

I believe most revisions should be done in the results and discussion section. The result section now only shows statistics for the found difference in recommendations of shared living, but there are more options to answer. For completion, it would be more transparent to also show the non-significant results. Especially because the discussion section does try to explain some of the non-findings, but no statistics are given. The result section should also include some descriptive statistics of the answers, as for example in the discussion the scored weakness of the parents is discussed, but no statistics are provided.

Response: We have added the numerical data on custody despite that only six participants in the whole study recommended sole custody in the entire sample. The study did not include any ratings of parental weakness. We have revised the sentence "However, the weaknesses were rated as more serious for fathers than mothers, and fathers were subsequently more excluded from the child to a larger extent compared to mothers." This sentence is confusing and is revised into "However, the identical weaknesses in both parents only had an effect on fathers, as they were recommended to be more excluded from the child's life to a larger extent compared to mothers."

Results and discussion section II

Furthermore, research on bias is important but before you can conclude that there is a bias, it is important to exclude other possible explanations of differences. Now, nothing is stated about any additional analyses trying to find other explanations such as gender of the participant or years of experience. If this is not possible with the sample size, the authors should consider trying to increase the sample size.

Response: As a consequence of adding group data in the design and participants section, there are now statistical analyses on group data that could have been confounding factors such as participant age or years of experience. We wholeheartedly agree that a larger sample size would be preferable. However, given that it is professionals we are trying to recruit, the acceptance rate is quite low. Also, one has to bear in mind that family court units are small and several of the smaller municipalities (of Sweden's 290) are using the same custody evaluators.

Results and discussion section III

In general, the result and discussion sections require more explanation about made choices for the analysis. The method section mentions four answer options, but the structure and text of the result section makes it difficult to follow which answer options are analyzed or taken together. The second paragraph of the discussion was very hard to follow, I believe the first sentence should be “… sole living but not sole custody…” instead of shared custody.

Response: The results are dependent on participants' recommendations. For instance, there is a floor effect, where only six participants recommended sole custody which is insufficient for statistical analysis. We have added a section explaining the intricate relation between custody, living and visitation recommendations to introduce our analysis. We have revised and simplified the wording to increase intelligibility in the second discussion section.

Results and discussion section IV

The discussion section discusses some possible explanations of the findings but does not elaborate on the non-finding for custody vs. the finding on living. It would strengthen the paper if the authors could come up with an explanation for this as this is somewhat surprising for me as a reader.

Response: We have added a section that discusses the most obvious explanation for this; case severity.

Minor thing

A minor thing, but I hope that there is more recent work on unrealistic vignette stories than from 200 years ago.

Response: Our initial reference concerned the first notion of finding a story (even fictional stories) non-realistic. We have removed this reference.

Reviewer #2

While the paper focuses on an important topic, it would be good to further examine the impact of your findings. What does this mean for practitioners? What needs to happen in the field to reduce the negative impact on men?

Response: Thank you for this wise comment. We added a section "Implications for practitioners" where we discuss what practitioners need to do as well as a section "Implications for children and fathers". However, it is rather limited because we are cautious about letting this discussion end up as a scientific manifesto of men's rights in the family courts or delving into research on youth criminality because of the lack of fathers or father-figures.

Reviewer #3

Method

It is unclear from the manuscript whether the risk question was asked before the living recommendation.

Response: Thank you for pointing this out. We clarified this in the procedure section.

Results I

I recommend reporting and analyzing the risk data here.

Response: As much as we appreciate the interest in these data, we have decided against this recommendation for various reasons. We plan to write another paper based on this data. Perhaps unfair to the reviewer who is unable to see the data. Our strong suspicion, however, is that presentation of the risk data would risk the entire paper would be derailed into a paper on low levels of inter-rater reliability, and the potential societal issues such as limited rule of law that presents. We are cautious about letting the important gender discrimination data and message drown in the ocean of risk data. We hope that this argument is acceptable, and that the interested reviewer has patience for our future manuscript.

Results II

The sample size is very small: a total of 29 Swedish custody evaluators (8 men and 21 women). Relatedly, was there an interaction between the gender of the decision-maker and the gender of the parent? I am concerned that the sample size is too small to investigate this adequately.

Response: This is why we have statistics, also very small sample sizes can be significant if the effect size is large enough, or stated differently in this case that gender discrimination against men is sufficiently severe. In this context, that we do find significant findings in a small data set is an interesting finding in itself. We agree that the sample is small, and the present sample cannot detect any differences in ratings between the male and female participants. Since we cannot do this analysis, we have added this as a natural suggestion for future studies.

Discussion I

The authors acknowledge the risk of additional discrimination occurring in the judicial court. However, it could also be that courts "correct" the discriminatory decisions of evaluators. This should also be stated as a limitation.

Response: This is a good point. We have added a discussion on this as a limitation along with some empirical arguments that the courts do not correct the custody evaluations.

Reviewer #4

Overall, the manuscript indicates the exciting point of 'discrimination against men in custody dispute'. However, it requires more details on the significance of the study, its methodology and results for meaningful discussions. The manuscript's introduction still lacks an explanation of the importance of gender discrimination in men's studies.

Response: Thank you for this comment. We have revised the manuscript in and added more background

General point

It is still being determined whether the research is quantitative or qualitative.

Response: Thank you for this comment. To address this, we have changed the title to "Gender Discrimination in Swedish Family Courts: A Quantitative Vignette study".

Results

As the quantitative research, the sampling technique should be demonstrated. From which population can a representative samples of 29 social workers be resembled?

Response. This comment is confusing to us. First, a small sample by itself, does not mean that the representativeness of the sample must be limited. Furthermore, there is a limited number of custody evaluators in Sweden. We recruited participants from several municipalities in the middle of Sweden, including Stockholm, which is the largest city in Sweden. We have added this information in the Design and Participants section.

Results II

As for the qualitative research, the data results are invalid for hypothesis testing. The demographic data of all social workers should be included in discussions. Previous work on gender discrimination with the vignette method demonstrates that both the gender and social capital of participants play a significant role as well.

Response: This is a very good point. However, the social worker profession is heavily dominated by women, and our sample, although small, represents the gender distribution very well.

Discussion

Instead of discussion on the 'weaknesses' part of the study, the study should close the gap in its methodology and provide more insightful findings for further discussions.

Response: We have revised the discussion section by adding more topics such as implications for practitioners, and implications for fathers and children. We hope that this is an adequate revision.

---

## [Decision Letter · Decision Letter 1]

19 Jun 2024

PONE-D-23-40878R1Gender Discrimination in Swedish Family Courts: A Quantitative Vignette StudyPLOS ONE

Dear Dr. Sikström,

Thank you for submitting your manuscript to PLOS ONE. After careful consideration, we feel that it has merit but does not fully meet PLOS ONE’s publication criteria as it currently stands. Therefore, we invite you to submit a revised version of the manuscript that addresses the points raised during the review process.

Specifically, we suggest the following:

Section 'The primary care-taker preference', Paragraph 2

- This paragraph needs to be rewritten in order to A) ensure that provided statements are supported by recent references and B) clarify how athletics is associated with male gender stereotypes or is relevant to the discussion, or preferably, remove it altogether.

Section, 'Significance of gender discrimination in child custody', Paragraph 1

- Rephrase what is meant by "This can only benefit women." (we have concerns about a potential statement of false equivalence being made here).

Section, 'Method'

- Provide information on statistical analysis performed

- Provide vignettes as SI file

Section Discussion, Paragraph 1

- "In terms of custody recommendations, results showed no evidence for gender discrimination" - this is not in line with the results section above - the results were shown to be inconclusive for this part of the study - this needs to updated accordingly, as well as potential removal of the discussion regarding the floor effect.

Section, Implications for fathers and children, Paragraph 1

- we suggest that the discussion and comparison against the job market is not relevant and should be removed.

We look forward to receiving your revised manuscript.

Kind regards,

Andrea Cioffi

Academic Editor

PLOS ONE

Journal Requirements:

Reviewers' comments:

Reviewer's Responses to Questions

**Comments to the Author**

1. If the authors have adequately addressed your comments raised in a previous round of review and you feel that this manuscript is now acceptable for publication, you may indicate that here to bypass the “Comments to the Author” section, enter your conflict of interest statement in the “Confidential to Editor” section, and submit your "Accept" recommendation.

Reviewer #1: (No Response)

Reviewer #2: All comments have been addressed

2. Is the manuscript technically sound, and do the data support the conclusions?

Reviewer #1: Partly

Reviewer #2: Yes

3. Has the statistical analysis been performed appropriately and rigorously? 

Reviewer #1: Yes

Reviewer #2: Yes

4. Have the authors made all data underlying the findings in their manuscript fully available?

Reviewer #1: Yes

Reviewer #2: Yes

5. Is the manuscript presented in an intelligible fashion and written in standard English?

Reviewer #1: Yes

Reviewer #2: Yes

6. Review Comments to the Author

Reviewer #1: I believe that the added information about gender equality does not benefit the paper; the data on itself is interesting enough and does not need to be compared with gender bias against women. I would highly recommend to keep this paper separate from the comments about that gender equality for women is a disadvantage for men. It would strengthen both the introduction and discussion to focus just on this type of gender bias instead of minimizing other types in order to show the relevance of the research.

Reviewer #2: The authors have responded to feedback well and in doing so have made their publication much stronger. I applaud them for delving into such a difficult topic, and encourage them to do more work in the future and not to be too scared off by the women's rights movements. All genders should have equal rights and in an ideal world, one persons rights should not breach another persons rights.

7. PLOS authors have the option to publish the peer review history of their article (what does this mean?). If published, this will include your full peer review and any attached files.

Reviewer #1: No

Reviewer #2: No

---

## [Author Response · Author response to Decision Letter 1]

28 Jun 2024

Dear editor/reviewer

We have made changes in PONE-D-23-40878R1 according to your suggestions as specified below.

Specifically, we suggest the following:

Section 'The primary care-taker preference', Paragraph 2

- This paragraph needs to be rewritten in order to A) ensure that provided statements are supported by recent references and B) clarify how athletics is associated with male gender stereotypes or is relevant to the discussion, or preferably, remove it altogether.

Our response: We have removed the examples given in this paragraph. The paragraph now reads:

“Historically and traditionally, women have done these tasks more than men. This is particularly true when the child is an infant or very young. However, the importance of fathers becomes more pronounced as the child grows older, are more ready to be introduced to adulthood and how to handle the outside world. However, there is a strong notion that mothers are the primary caregiver in many cultures. As Warshak (1996) points out, there are several care-taking activities associated with traditional female gender stereotypes that are focused on in custody disputes. In contrast, such activities are typically not associated with male gender stereotypes. Similar to the tender years doctrine, the notion of the primary care-taker favored mothers over fathers.” 

Section, 'Significance of gender discrimination in child custody', Paragraph 1

- Rephrase what is meant by "This can only benefit women." (we have concerns about a potential statement of false equivalence being made here).

Our response: This statement has been removed. 

Section, 'Method'

- Provide information on statistical analysis performed

Our response: The missing inference test in the first paragraph of the results section has now been added. 

- Provide vignettes as SI file

Out response: The Vignettes SI file labeled “NgaosuvanVignettes.pdf” has been attached to the submission 

- "In terms of custody recommendations, results showed no evidence for gender discrimination" - this is not in line with the results section above - the results were shown to be inconclusive for this part of the study - this needs to updated accordingly, as well as potential removal of the discussion regarding the floor effect.

Our response: This has been removed: “ In terms of custody recommendations, results showed no evidence for gender discrimination. This fact that this variable was not significant is likely due to a floor effect in the data, where only six participants recommended sole custody. “

Section, Implications for fathers and children, Paragraph 1

- we suggest that the discussion and comparison against the job market is not relevant and should be removed.

Our response: This has been removed: “This is not true for discrimination in the job market, where a person discriminated against on a job application, can apply to other jobs. Fathers losing custody disputes have no such opportunity”. 

Reviewer #1: I believe that the added information about gender equality does not benefit the paper; the data on itself is interesting enough and does not need to be compared with gender bias against women. I would highly recommend to keep this paper separate from the comments about that gender equality for women is a disadvantage for men. It would strengthen both the introduction and discussion to focus just on this type of gender bias instead of minimizing other types in order to show the relevance of the research.

Our response: We have deemphasized the gender equality comparisons in the manuscript.

Reviewer #2: The authors have responded to feedback well and in doing so have made their publication much stronger. I applaud them for delving into such a difficult topic, and encourage them to do more work in the future and not to be too scared off by the women's rights movements. All genders should have equal rights and in an ideal world, one persons rights should not breach another persons rights.

Our response: Thank you.

---

## [Editor Report · Decision Letter 2]

5 Jul 2024

Gender Discrimination in Swedish Family Courts: A Quantitative Vignette Study

PONE-D-23-40878R2

Dear Dr. Sikström,

We’re pleased to inform you that your manuscript has been judged scientifically suitable for publication and will be formally accepted for publication once it meets all outstanding technical requirements.

Kind regards,

Andrea Cioffi

Academic Editor

PLOS ONE
---

## [Editor Report · Acceptance letter]

10 Jul 2024

PONE-D-23-40878R2 

PLOS ONE

Dear Dr. Sikström, 

I'm pleased to inform you that your manuscript has been deemed suitable for publication in PLOS ONE. Congratulations! Your manuscript is now being handed over to our production team.

Kind regards, 

on behalf of

Dr. Andrea Cioffi 

Academic Editor

PLOS ONE